# 3D Liver and Tumor Segmentation with CNNs Based on Region and Distance Metrics

**Yi Zhang [1], Xiwen Pan [2], Congsheng Li [1] and Tongning Wu [1,]***

[1] China Academy of Information and Communications Technology, Beijing 100191, China; zhangyi5@caict.ac.cn (Y.Z.); licongsheng@caict.ac.cn (C.L.)

[2] Beijing University of Posts and Telecommunications, Beijing 100876, China; panxiwen@bupt.edu.cn

[*] Correspondence: wutongning@caict.ac.cn; Tel.: +86-010-62300217

**Abstract:** Liver and liver tumor segmentation based on abdomen computed tomography (CT) images is an essential step in computer-assisted clinical interventions. However, liver and tumor segmentation remains the difficult issue in the medical image processing field, which is ascribed to the anatomical complexity of the liver and the poor demarcation between the liver and other nearby organs on the image. The existing 3D automatic liver and tumor segmentation algorithms based on full convolutional networks, such as V-net, have utilized the loss functions on the basis of integration (summing) over a segmented region (like Dice or cross-entropy). Unfortunately, the number of foreground and background voxels is usually highly imbalanced in liver and tumor segmentation tasks. This greatly varies the value of regional loss between various segmentation classes, and affects the training stability and effect. In the present study, an improved V-net algorithm was applied for 3D liver and tumor segmentation based on region and distance metrics. The distance metric-based loss function utilized a distance metric of the contour (or shape) space rather than the area. The model was jointly trained by the original regional loss and the three distance-based loss functions (including Boundary (BD) loss, Hausdorff (HD) loss, and Signed Distance Map (SDM) loss) to solve the problem of the highly unbalanced liver and tumor segmentation. Besides, the algorithm was tested in two databases LiTS 2017 (Technical University of Munich, Munich, Germany, 2017) and 3D-IRCADb (Research Institute against Digestive Cancer, Strasbourg Cedex, France, 2009), and the results proved the effectiveness of improvement.

**Keywords:** liver and tumor segmentation; full convolutional networks; distance metrics

## 1. Introduction

Liver together with related lesion automatic segmentation represents a vital link to obtain quantitative biomarkers for the support systems to accurately diagnose in the clinic and make decisions based on the computer [1]. Nonetheless, liver segmentation remains a challenge in the medical image processing field, which is due to the anatomical complexity of the liver and the poor demarcation between the liver and other neighboring organs [2]. Accurate measurements based on the computed tomography (CT) image, such as location, shape, and the volume of the tumor, together with the functional liver volume, helps physicians evaluate hepatocellular carcinoma (HCC) and plan treatment [3]. However, manually outlining the target organ on every slice can be greatly demanding and effort-consuming; besides, the obtained results are subjective [1].

Two grand challenges benchmarks were carried out with the coordination of the MICCAI (Medical Image Computing and Computer-Assisted Intervention Society) conference to segment the liver and related lesions in 2007 and 2008, respectively [4,5]. Several approaches based on artificial design features were proposed for liver and related lesion segmentation base on CT images.

Recently, thresholding [6,7], graph cut and level set techniques [8–10], region growing, and deformable model-based methods [11,12] have been applied in research to segment the liver and related lesions. However, those methods require substantial human intervention, which may cause bias and mistakes. Therefore, it is necessary to develop automatic and end-to-end approaches to segment tumors on CT images [13].

The scientific community has paid great attention to deep Convolutional Neural Networks (CNN) to solve tasks in computer vision, including recognizing, classifying, and segmenting objects [14–17]. Similarly, novel deep learning-based segmentation approaches have been put forward to analyze medical images, which achieve greatly competitive findings in comparison with state-of-the-art methods [18–22]. End-to-end CNN-based approaches have been verified to be sound for analyzing image appearance, and this has motivated researchers to employ them for fully automatic segmentation of the liver and related lesions within the CT volumes.

The existing deep learning-based liver and tumor segmentation studies are roughly divided into two classes—(1) 2D Fully Convolutional Networks (FCN), like U-Net [23], multi-channel FCN [24], as well as VGG (Oxford Visual Geometry Group) based FCN [25]; (2) 3D FCN, in which 2D convolutions are substituted with the 3D convolutions in the presence of volumetric input data [26,27].

The 2D FCN-based methods use 2D slices from 3D volumes for the segmentation task. Specifically, singular or three neighboring slices cropped based on volumetric images are incorporated into the 2D FCNs [24,25], and then 2D segmentation maps are stacked to produce a segmentation volume. Sun et al. [24] designed a multi-channel fully convolutional network (MC-FCN) to segment liver tumors from multi-phase contrast-enhanced CT images. Since each stage of the contrast-enhanced data provided information about pathological features, it is possible to generate fusion feature maps through merging features from different channels. However, the spatial structural organizations of organs are not considered, and the volumetric information is not fully explored. It remains insufficient to explore spatial data even though neighboring slices are used, which may degrade the performance of the segmentation [3].

The 3D FCN-based method can avoid discontinuities between adjacent slices. For instance, Özgün Çiçek et al. introduced a 3D U-Net network for volumetric segmentation that learns from sparsely annotated volumetric images [28]. The network extended the previous U-Net [18] architecture by replacing all 2D operations with their 3D counterparts. The implementation performed on-the-fly elastic deformations for efficient data augmentation during training. Milletari et al. [29] had put forward the V-net architecture, the U-Net 3D variant, to segment 3D images by the direct use of 3D convolutional layers that had an objective function on the basis of the Dice coefficient. A suitable loss function is an important measure for the usefulness of segmentation for an intended task. Nonetheless, the widely used loss functions for 3D FCNs, including cross-entropy or Dice, are proposed on the basis of integrals (summations) in various segmentation volumes. Besides, the regional loss in segmentation with a high imbalance level (which is common in liver and tumor segmentation) leads to substantially different values among different segmentation types, and this possibly affects the training stability and performance. Kervadec et al. [30] proposed the boundary loss concept, where contour or shape space, rather than regions, was used to form the distance metric. It reduced the regional loss difficulty when there were problems of substantially imbalanced segmentation, since integrals were used along the inter-regional boundary (interface) rather than the imbalanced integrals across regions.

Based on similar ideas, Karimi et al. [31] proposed Hausdorff (HD) loss based on distance metrics. Although HD was utilized to evaluate the performance of image segmentation algorithms, those algorithms rarely aimed at minimizing HD directly. An "HD-Inspired" loss function was proposed in [31] that could be used for a stable training segmentation model with the goal of reducing HD directly.

Moreover, Xue et al. [32] proposed a distance-based loss function named the Signed Distance Map (SDM) loss. Due to the rigorous mapping of the signed distance map computed based on the object boundary contour with a binary segmentation map, they took advantage of learning SDM on the basis of the medical scans directly. The task of segmentation was converted into predictive SDM in

their method, which retained excellent segmentation performance and had better smoothness and shape continuity.

In this paper, an improved V-Net based on distance metric was utilized for the 3D liver and tumor segmentation tasks. Three distance-based loss functions, including BD loss, HD Loss, and SDM Loss, were used in combination with the original V-Net loss, respectively, to solve the problem of a highly unbalanced liver and tumor segmentation. In addition, the algorithm was tested on two databases, and the results proved the effectiveness of improvement.

## 2. Materials and Methods

### 2.1. Overall Framework

In this paper, we trained the 3D V-Net using the above-mentioned three distance-based loss functions and the regional loss function jointly. Figure 1 shows the 3D liver and tumor segmentation framework utilized in this paper.

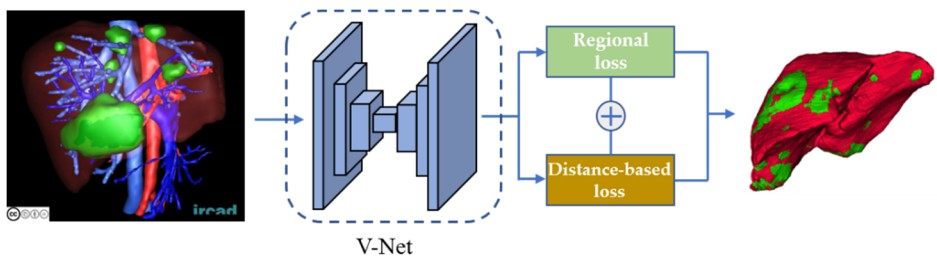

**Figure 1.** The liver and tumor segmentation framework.

In the training stage, the 3D data was fed into the V-net model for feature extraction, then the regional and the three distance-based loss functions (which were denoted as $Loss_{Reg}$, $Loss_{BD}$, $Loss_{HD}$, and $Loss_{SDM}$) were combined through variable weight values and used to jointly train the liver and tumor segmentation model. The loss functions in this paper were denoted as:

$$Loss_{BD} = \alpha Loss_{Reg} + (1-\alpha)Loss_{Boundary} \tag{1}$$

$$Loss_{HD} = \alpha Loss_{Reg} + (1-\alpha)\text{Loss}_{Hausdorff} \tag{2}$$

$$Loss_{SDM} = \alpha Loss_{Reg} + (1-\alpha)\text{Loss}_{Signed\ Distance\ Map} \tag{3}$$

where $Loss_{Reg}$ was the regional loss function utilized in the original V-net architecture, these loss functions will be described in detail in the next paragraph. Among them, the distance-based loss function was used to assist the regional loss function to fine-tune the training models. Therefore, at the beginning of training, $\alpha$ was set to 1, which indicated that the models were trained only utilizing regional loss, and the distance-based losses were not involved in the calculation of the loss function. When the training reached the plateau, the $\alpha$ gradually decreased until it reached a value of 0.01. As found in these experiments, such a training strategy was more effective than a joint training strategy from scratch. In the next paragraph, we will expand on the details of the algorithm.

### 2.2. Related Work

- V-Net

The 3D V-Net architecture, which was the U-Net 3D variant, was put forward to segment the 3D images. The model was proposed to train end-to-end Magnetic Resonance Imaging (MRI) volumes that depicted prostate, which also immediately learned to estimate whole volume segmentation [29]. Another important contribution of this work was the introduction of a new loss layer for segmentation tasks on the basis of the Dice coefficient, which was a commonly used regional overlap measure to

analyze medical images [33]. The schematic representation of the V-Net architecture is provided in Figure 2.

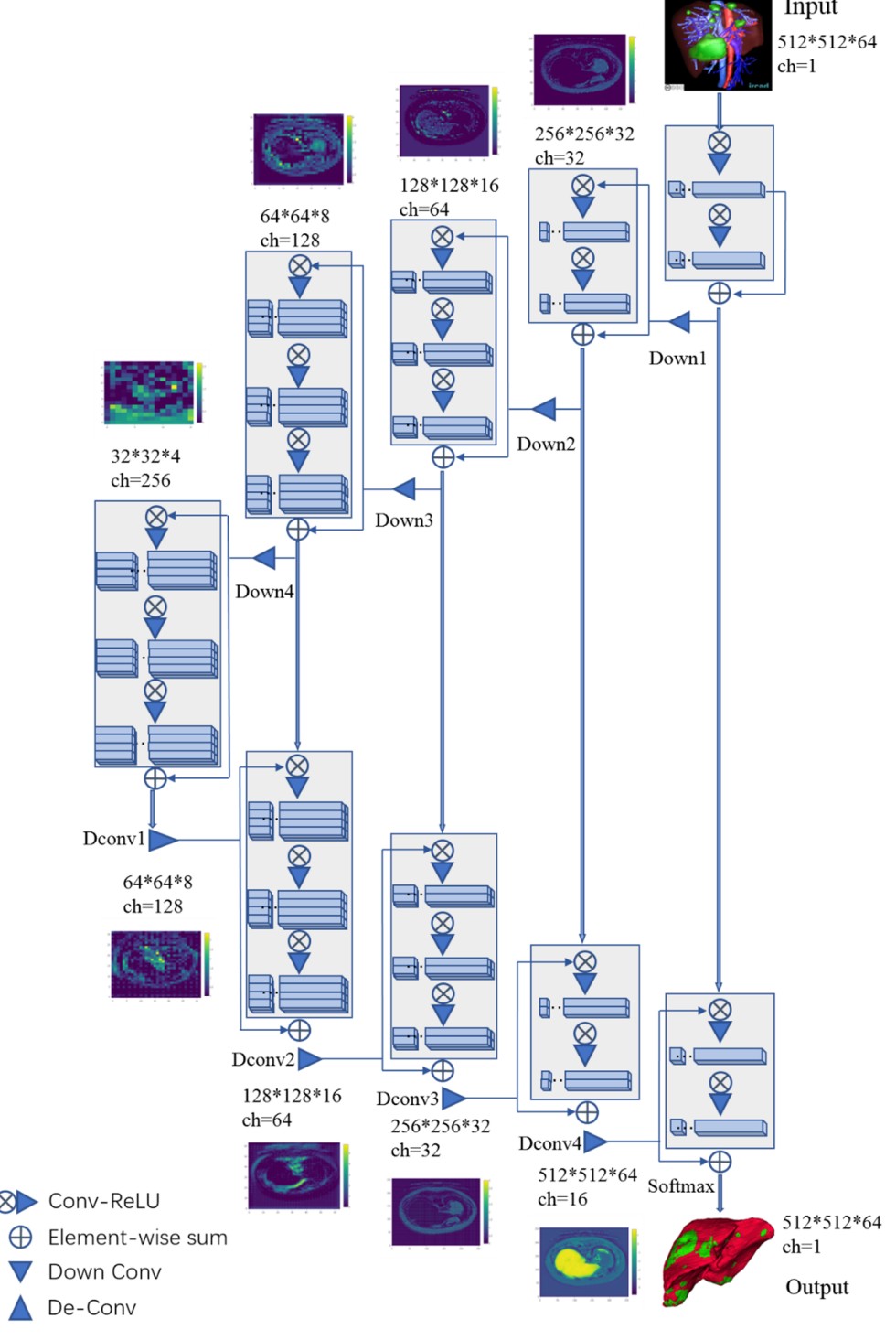

**Figure 2.** The schematic representation of the V-Net architecture.

The input of the 3D V-Net was 3D data. The first half of the network was constituted by the compression path, while the latter half decompressed the signals till they reached the size of their initial input data. The first half of the network was segmented to diverse phases according to distinct resolutions. Each phase contained one to three convolutional layers. The Parametric Rectified Linear

Unit (PReLU) was applied in the entire network. The latter half network extracted features and expanded spatial support for feature maps with low resolution. The deconvolution manipulation was performed to enhance the input size following every stage. In addition, this network also connected features collected via the early stages from the compression to the decompression side of the CNN. Thus, the fine-grained details were collected, which improved the eventual contour estimation quality. The regional loss functions, including cross-entropy and Dice loss, were jointly utilized in the V-net, which was denoted as:

$$Loss_{Reg} = Loss_{seg} + Loss_{Dice} \tag{4}$$

where $Loss_{seg}$ was the cross-entropy loss function and the $Loss_{Dice}$ was the Dice loss function.

- Cross-entropy loss

The voxel-wise cross-entropy loss was one of the most commonly used loss functions for image segmentation task. This loss will examine each voxel individually and compare the class prediction vector with the ground truth vector. The cross-entropy loss function is denoted as:

$$Loss_{seg}(p, g) = -\frac{1}{n} \sum_{i=1}^{N} [g_i \log p_i + (1 - g_i) \log(1 - p_i)] \tag{5}$$

where $p_i$ represents the probability that voxel $i$ belongs to the foreground, and $g_i$ represents the ground truth.

- Dice loss

A novel objective function on the basis of the Dice coefficient (range, 0–1) was utilized in V-Net [29]. The liver and tumor segmentation is a binary segmentation task, in which the soft-max layer outputs the probability that each voxel belongs to the foreground or background. For the Dice coefficient $D$ of two binary volumes, it is calculated by the following formula:

$$D = \frac{2 \sum_i^N p_i g_i}{\sum_i^N p_i^2 + \sum_i^N g_i^2} \tag{6}$$

In the formula, the sums run across N voxels in the estimated binary segmentation volume $p_i \in P$, together with the ground truth binary volume $g_i \in G$. The above formula to calculate Dice is also differentiated in terms of the prediction *j-th* voxel, yielding the gradient:

$$\frac{\partial D}{\partial p_j} = 2 \left[ \frac{g_j \left( \sum_i^N p_i^2 + \sum_i^N g_i^2 \right) - 2p_j \left( \sum_i^N p_i g_i \right)}{\left( \sum_i^N p_i^2 + \sum_i^N g_i^2 \right)^2} \right] \tag{7}$$

Using this loss layer, it was no longer necessary to assign the loss weights to different classes of samples during the training phase. Furthermore, the Dice loss function calculation formula can be used for both 2d and 3d data.

- Boundary loss

In [30], Kervadec et al. proposed a boundary loss, in which the distance of contour (or shapes) space, rather than regions, was measured. Boundary loss contributes to alleviating issues associated with regional loss in the context of substantially imbalanced segmentation tasks. In addition, boundary loss also provided complementary data to those of regional loss. A symmetric *L2* distance (Euclidean distance) on the space of shapes (or contours) was expressed as a regional integral, which avoids completely local differential computations involving contour points. The non-symmetric *L2* loss function for regularizing segmentation mask S's boundary deviation compared with the ground truth G is written as follows:

$$Dist(p, g) = \int_G \|p_i - g_i\|^2 dg \tag{8}$$

In the formula, on the ground-truth boundary $G$, the boundary point $g_i$ is aligned based on the counterpart $p_i$ that is located on $P$ (the prediction boundary). The boundary loss function was used to segment the Magnetic Resonance (MR) images of the brain lesion in [30], and the Dice and Hausdorff score increased by 8% and 10%, respectively, relative to the baseline levels in which the generalized Dice was utilized to be the loss function [30].

- Hausdorff Loss

In [31], Karimi et al. put forward a loss function based on the direct HD reduction for training the CNN-based segmentation algorithms. They proposed three approaches for estimating the HD based on the map of segmentation probability. One method was to use a distance transform splitting of the boundary. The second approach was developed on the basis of the use of morphological erosion to those differences in the real segmentation maps compared with estimated counterparts. The last approach was to employ spherical convolution kernels with diverse radii to the map of the segmentation probability. According to the above three approaches proposed to estimate HD, three loss functions were also put forward in training for the sake of HD reduction. Karimi et al. had optimized the function on the basis of the Hausdorff distance to compare the estimated segmentation with the ground truth one, which is shown below.

$$f_{HD}(p,g) = Loss(p,g) + \lambda\left(1 - \frac{2\sum_\Omega(p \circ g)}{(p^2 + g)}\right) \tag{9}$$

In the formula, the second term is the Dice loss function, whereas the first one is the Hausdorff distance of $p$ and $g$. Parameter $\lambda$ is the ratio of the HD-based loss term to the Dice loss term. Let $\Omega$ denote the grid on which the image I is defined, and $p$ and $g$ denote the segmentation and ground truth, respectively. The predicted and ground truth segmentation, separately:

$$Loss(p,g) = \frac{1}{|\Omega|}\sum_\Omega\left((p-g)^2 \circ \left(d_p^\alpha + d_g^\alpha\right)\right) \tag{10}$$

Parameter $\alpha$ determines the penalty level for a large error. $d_g$ stands for the ground truth segmentation distance map, which represents an unsigned distance to the $\delta g$ boundary. Similarly, represents the distance to $\delta p$. Meanwhile, indicates the Hadamard operation. In this paper, the HD loss function was utilized for training the V-net model jointly with the regional loss function.

- Signed Distance Map Loss

Xue et al. [32] put forward a novel algorithm for solving those existing problems in the current organ segmentation systems based on deep learning. These systems frequently generated results that were unable to obtain target organ shape, together with a lack of smoothness. The task of segmentation was converted into predictive SDM in their method, since there was a rigorous mapping between the SDM and the binary segmentation map. For the target organ, as well as a point $x$ shown on the 3D medical image, $y$ is the most adjacent point on the target organ surface, the SDM, which maps $R$ to $R^3$ can be deemed below:

$$\Phi(x) = \begin{cases} 0, & x \in S \\ -\inf_{y \in S}\|x-y\|_2, & x \in \Omega_{in} \\ +\inf_{y \in S}\|x-y\|_2, & x \in \Omega_{out} \end{cases} \tag{11}$$

In the formula, $S$ represented the target organ surface; $\Omega_{in}$ together with $\Omega_{out}$ denoted the target organ interior and exterior, separately. That was to say, the absolute SDM value indicated the distance between a specific point to the most adjacent one on the surface of the organ, whereas the sign indicated

the organ interior or exterior. Notably, the zero level or zero distance set indicates the presence of the point on the organ surface. The SDM loss is defined as:

$$L_{SDM} = L_1 - \sum_{t=1}^{C} \frac{g_t p_t}{(g_t p_t + p_t^2 + g_t^2)} \tag{12}$$

where $L_1$ represented the $L_1$ loss, which is the $L_1$ difference between the predicted and the real SDM values. $g_t$ represents the ground truth SDM, and $p_t$ denotes the predicted SDM.

## 3. Experiment and Discussion

### 3.1. Experimental Preparation and Protocols

We conducted an evaluation of two datasets for liver and tumor segmentation, and the examples of the data are shown in Figure 3.

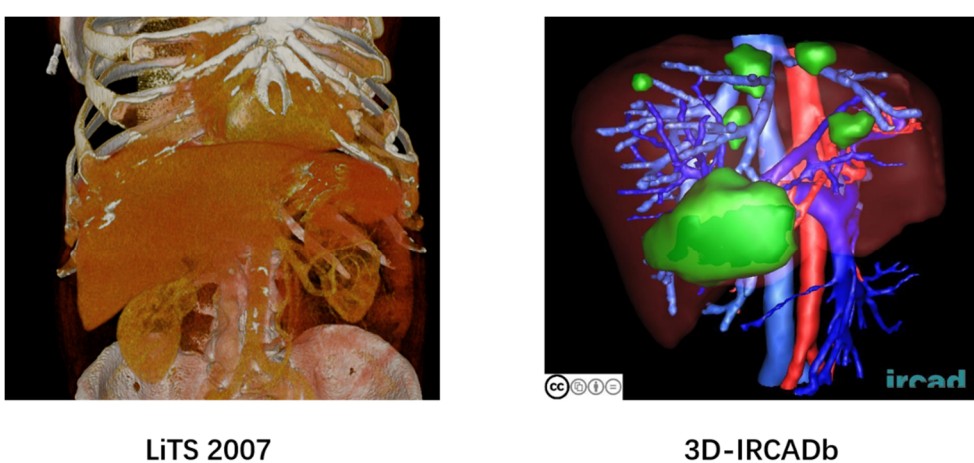

**LiTS 2007**　　　　　　　　**3D-IRCADb**

**Figure 3.** The scans in Liver Tumor Segmentation Challenge (LiTS) 2017 dataset and 3D Image Reconstruction for Comparison of Algorithm Database (3D-IRCADb).

LiTS 2017: Liver Tumor Segmentation Challenge (LiTS) dataset [34] provides 201 contrast-enhanced 3D abdominal CT scans, and segmentation labels for liver and tumor regions with a resolution of $512 \times 512$ in each axial slice. There are 131 scans providing ground-truth labels, and 70 scans that do not provide labels. The in-plane resolution ranges from 0.60 mm to 0.98 mm, and the slice spacing from 0.45 mm to 5.0 mm. We clipped the intensity values to the range [−300, 400] HU to ignored irrelevant details and normalized the images into [0, 1].

3D-IRCADb: 3D Image Reconstruction for Comparison of Algorithm Database (3D-IRCADb) is a database containing anonymous medical images of several groups of patients, as well as manual segmentation of various structures of interest by clinical experts. The database consists of 3D CT scans of 10 women and 10 men with liver tumors. The in-plane resolution ranges from 0.57 mm to 0.87 mm, and the slice spacing from 1.6 mm to 4.0 mm. All scans were performed while the arterial phase was in the inhaled position.

We trained the four 3D V-Net models with regional loss and other three distance-based loss functions in two databases. The experiments were conducted with Torch and optimized with the Adam algorithm [35] on four NVIDIA Tesla V100 GPUs. (Gigabyte Technology, Beijing, China) When the deep models were trained, a batch size of 2 and a learning rate of 0.001 were employed. Of them, the learning rate was divided by five after 200 epochs, and the training ended after 1400 epochs. For fairly comparing the diverse loss functions, the models were tested on the test set after every 40 epochs, and the best models were utilized for the contrast test.

As mentioned above, the value of $\alpha$ in Equations (1)–(3) was set to 1 from the initial training to 400 epochs. Afterward, it was reduced by 0.01 every 10 epochs till reaching 0.01. Therefore, only the regional loss was used for training at the beginning, and then the distance-based loss influence gradually increased. According to our results, the as-proposed convenient scheduling strategy always gave superior results over the constant value.

*3.2. Experimental Results*

3.2.1. Quantitative Evaluation

In the LiTS 2017 dataset, 131 scans were used as the experimental data, among which, 102 were used for training, 20 for verification, and nine for testing. In the 3D-IRCADb dataset, 20 scans were selected, including 10 utilized as the training set, five as the validation set, and five as the test set. Three models with HD loss, BD loss, and SDM loss functions were trained on these two datasets, respectively. We utilized the values of Dice Similarity Coefficient (DSC), the 95th percentile of the Hausdorff Distance (HD) metrics (HD95), the average symmetric surface distance (ASD), together with the True Negative Rate (TNR, specificity) and True Positive Rate (TPR, sensitivity) as the evaluation indicators. The definitions of the indicators were shown as follows:

$$DSC = \frac{2|TP|}{2|TP| + |FN| + |FP|} \tag{13}$$

where TP, TN, FP, and FN represent the true positive, true negative, false positive, and negative values, respectively. The HD95 was defined as the 95$^{\text{th}}$ percentile of the Hausdorff distance between the predicted delineation and the ground truth annotation, which was a common indicator in image segmentation tasks.

If $S(A)$ denotes the set of surface voxels of $A$, then the shortest distance of an arbitrary voxel $v$ to $S(A)$ is defined as:

$$d(v, S(A)) = \min_{s_A \in S(A)} \|v - s_A\| \tag{14}$$

where $\|.\|$ denotes the Euclidean distance. The other indicators were calculated according to the following Equation.

$$ASD(A, B) = \frac{1}{|S(A)| + |S(B)|} \left( \sum_{s_A \in S(A)} d(s_A, S(B)) + \sum_{s_B \in S(B)} d(s_B, S(A)) \right) \tag{15}$$

$$TNR = \frac{|TN|}{|TN| + |FP|} \tag{16}$$

$$TPR = \frac{|TP|}{|TP| + |FN|} \tag{17}$$

Note that in the previous studies, researchers used data with different resolutions for training and testing, such as $512 \times 512$ [36], $256 \times 256$ [13], $224 \times 224$ [3], and $160 \times 160$ [26]. Therefore, we firstly trained the original v-net models with different sized data to evaluate the accuracy and computational efficiency. Table 1 summarized the DSC values and runtime of the liver and tumor segmentation task of three original V-net models trained utilizing different resolution data ($512 \times 512$, $256 \times 256$, and $128 \times 128$), respectively on two databases. The experimental results showed that the model trained with $512 \times 512$ resolution data has a segmentation result improvement of about 2% for the liver and 7% for the tumor compared to the model trained with low resolution. Especially for the tumor segmentation task with a smaller target, down sampling has a greater negative influence. Therefore, we tend to train and test the models utilizing the data with $512 \times 512$ resolution.

**Table 1.** Comparison of the results of models trained with different resolution data.

| Category | Dataset | Resolution | Dice Similarity Coefficient (DSC) |
|---|---|---|---|
| Liver | LiTS 2017 | 512*512 | **0.953** |
| | | 256*256 | 0.947 |
| | | 128*128 | 0.936 |
| | 3D-IRCADb | 512*512 | **0.929** |
| | | 256*256 | 0.924 |
| | | 128*128 | 0.91 |
| Tumor | LiTS 2017 | 512*512 | **0.699** |
| | | 256*256 | 0.655 |
| | | 128*128 | 0.615 |
| | 3D-IRCADb | 512*512 | **0.623** |
| | | 256*256 | 0.597 |
| | | 128*128 | 0.567 |

Table 2 summarizes the results obtained from the LiTS 2017 and 3D-IRCADb datasets. Compared to the models trained using only the region-based loss function ($L_{Reg}$), the segmentation results were improved by utilizing the joint loss functions ($L_{HD}$, $L_{BD}$, and $L_{SDM}$), which were evidenced by the indicators. The best results are shown in bold. For liver segmentation and tumor segmentation tasks, the distance-based loss functions improved the DSC by about 1.2% and 6.5% on the LiTS 2017 dataset, and they improved the DSC coefficients by 1.9% and 5.9% on the 3D-IRCADb dataset, respectively. On the LiTS 2017 test set, the HD95 reduced by 40.6% and 28.2%, while it decreased by 52.5% and 29.6% on the 3D-IRCADb test set. As for ASD, it decreased by 45.3% and 42.4% on the LiTS 2017 test set, and the decrease on the 3D-IRCADb dataset were 29.8% and 24.2%, respectively.

**Table 2.** A summary of the results on LiTS 2017 and 3D-IRCADb datasets.

| Category | Dataset | Loss Function | DSC | 95th Percentile of the Hausdorff Distance Metrics (HD95) (mm) | Average Symmetric Surface Distance (ASD) (mm) | True Negative Rate (TNR) | True Positive Rate (TPR) |
|---|---|---|---|---|---|---|---|
| Liver | LiTS 2017 | $L_{Reg}$ | 0.953 | 5.44 | 1.61 | 0.957 | 0.998 |
| | | $L_{HD}$ | 0.962 | 3.60 | 1.05 | 0.971 | 0.998 |
| | | $L_{BD}$ | 0.963 | 4.24 | **0.88** | 0.973 | 0.999 |
| | | $L_{SDM}$ | **0.965** | **3.23** | 1.07 | **0.982** | **0.999** |
| | 3D-IRCADb | $L_{Reg}$ | 0.929 | 8.74 | 2.58 | 0.921 | 0.997 |
| | | $L_{HD}$ | 0.942 | 6.97 | 2.17 | 0.949 | 0.998 |
| | | $L_{BD}$ | 0.947 | **4.15** | 1.87 | 0.955 | 0.998 |
| | | $L_{SDM}$ | **0.948** | 4.68 | **1.81** | **0.958** | 0.998 |
| Tumor | LiTS 2017 | $L_{Reg}$ | 0.699 | 9.36 | 2.17 | 0.655 | 0.998 |
| | | $L_{HD}$ | 0.731 | 8.77 | 1.82 | 0.682 | 0.999 |
| | | $L_{BD}$ | 0.745 | 8.14 | 1.68 | 0.708 | 0.999 |
| | | $L_{SDM}$ | **0.764** | **6.72** | **1.25** | **0.761** | **0.999** |
| | 3D-IRCADb | $L_{Reg}$ | 0.623 | 13.46 | 3.72 | 0.564 | 0.999 |
| | | $L_{HD}$ | 0.648 | 11.25 | 3.08 | 0.587 | 0.999 |
| | | $L_{BD}$ | 0.677 | 9.88 | 2.88 | **0.674** | 0.999 |
| | | $L_{SDM}$ | **0.682** | **9.47** | **2.82** | 0.654 | **0.999** |

Also, those distance-based loss functions improved the TNR of the liver segmentation and tumor segmentation tasks on the test sets. The improvement was 2.5% and 10.6% on the LiTS 2017 dataset, and 3.7% and 11.0% on the 3D-IRCADb dataset. It was also found that the TPR of all models has reached more than 99.7%, which shows that for the liver segmentation task, the existing models have already obtained high sensitivity [37].

The models on the 3D-IRCADb database didn't perform as well as it did on the LiTS 2017 database. It was mainly due to the small scale of the 3D-IRCADb database, resulting in insufficient model training.

Some studies have shown that utilizing additional data for training will significantly improve the performance of the 3D-IRCADb database [8,9,24,27].

Generally, the traditional region-based segmentation methods measure the affinity of the network probability Softmax output-defined region to related ground truth regions. It assumes that all samples and classes have the same importance distribution. Therefore, it requires a training set with balanced classes to get good generality. However, for unbalanced data, the regional loss-based approaches lead to training instability and biased decision boundaries to most categories.

Adding a distance-based loss function to the regional loss function for joint training can mitigate the issues. Instead of utilizing the imbalanced integrals on these regions, the distance-based loss function used integrals on the inter-regional boundaries. Therefore, it was easily combined with a regional loss for joint training to solve the problem of imbalanced data for the liver and tumor segmentation task.

In addition, the cross-validation experiments were also conducted on the LiTS 2017 and 3D-IRCADb datasets. The models trained utilizing the LiTS 2017 training set were tested on the 3D-IRCADb testing set, meanwhile, the 3D-IRCADb datasets trained models were also tested on the LiTS 2017 testing set. The experimental results in Table 3 show that the LiTS 2017 testing results of the models trained with the 3D-IRCADb training set were slightly decreased due to the relatively few training samples. The models trained on LiTS 2017 training set have achieved impressive testing results on the 3D-IRCADb testing set, which have hardly declined. In conclusion, the generalization ability of our algorithm has been proved.

**Table 3.** The cross-validation experimental results of the LiTS 2017 and 3D-IRCADb datasets.

| Category | Training Dataset | Testing Dataset | Loss Function | DSC | HD95 (mm) | ASD (mm) | TNR | TPR |
|---|---|---|---|---|---|---|---|---|
| Liver | 3D-IRCADb | LiTS 2017 | $L_{Reg}$ | 0.913 | 9.8 | 2.64 | 0.879 | 0.995 |
| | | | $L_{HD}$ | **0.924** | 8.06 | 2.25 | 0.914 | 0.996 |
| | | | $L_{BD}$ | 0.921 | 8.41 | 2.11 | 0.922 | 0.996 |
| | | | $L_{SDM}$ | 0.923 | **7.82** | **2.03** | **0.941** | **0.997** |
| | LiTS 2017 | 3D-IRCADb | $L_{Reg}$ | 0.919 | 10.74 | 3.01 | 0.858 | 0.995 |
| | | | $L_{HD}$ | 0.928 | 7.45 | 2.54 | 0.945 | 0.997 |
| | | | $L_{BD}$ | 0.926 | 7.12 | 2.75 | 0.951 | 0.997 |
| | | | $L_{SDM}$ | **0.934** | **6.88** | **2.31** | **0.960** | **0.998** |
| Tumor | 3D-IRCADb | LiTS 2017 | $L_{Reg}$ | 0.598 | 15.74 | 4.11 | 0.564 | 0.998 |
| | | | $L_{HD}$ | 0.644 | 12.32 | 3.77 | 0.58 | 0.998 |
| | | | $L_{BD}$ | **0.653** | 11.72 | **3.14** | 0.644 | 0.998 |
| | | | $L_{SDM}$ | 0.651 | **10.41** | 3.18 | **0.682** | **0.998** |
| | LiTS 2017 | 3D-IRCADb | $L_{Reg}$ | 0.587 | 18.22 | 4.62 | 0.526 | 0.999 |
| | | | $L_{HD}$ | 0.627 | 15.41 | 4.28 | 0.557 | 0.999 |
| | | | $L_{BD}$ | 0.631 | 13.64 | 3.62 | **0.682** | 0.999 |
| | | | $L_{SDM}$ | **0.634** | **13.25** | **3.24** | 0.674 | **0.999** |

### 3.2.2. Qualitative Evaluation

The qualitative results are shown in Figure 4. After visual inspection of the above results, great improvements were found when employing the distance-based loss function. Especially in the case of highly imbalanced between foreground and background voxels (such as rows 2, 4, and 5), the results of the joint training models have been greatly improved. The results obtained by the model only using the regional loss function showed that many small regions (including liver or tumor) were not correctly segmented. In contrast, adding distance-based loss functions for joint training improved the segmentation results by varying degrees. Furthermore, the model adding the SDM loss function obtained better results in most cases, which was also reflected in Table 2.

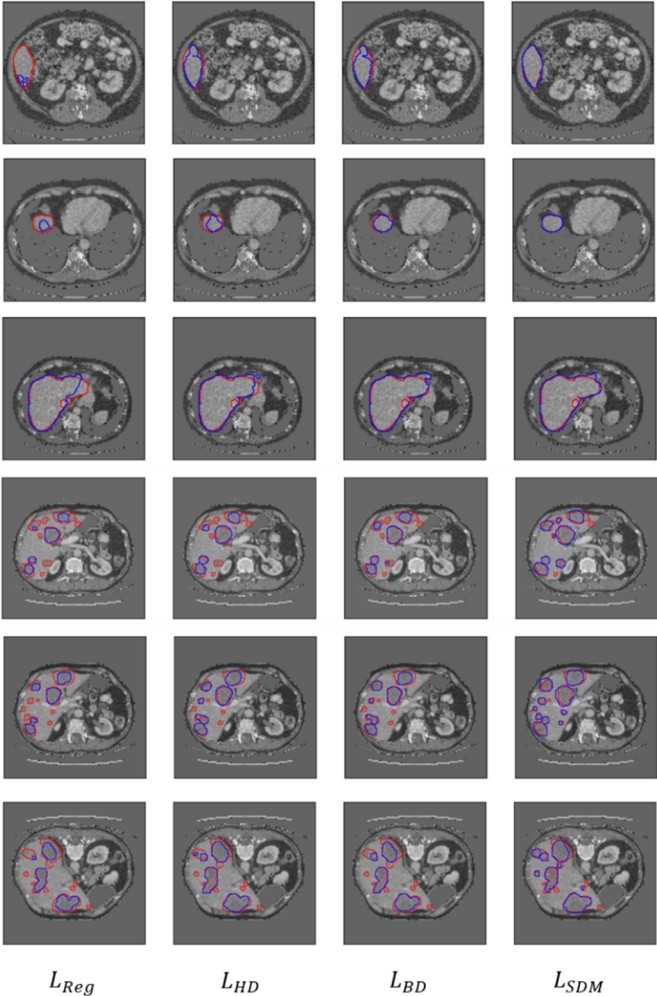

$L_{Reg}$       $L_{HD}$       $L_{BD}$       $L_{SDM}$

**Figure 4.** The visual comparison of segmentation results. The ground-truths are denoted in red, and the results are in blue. The first three rows are the results of liver segmentation; the last three rows are the results of tumor segmentation. Each column represents the results obtained, utilizing different loss functions.

## 4. Comparison and Discussion

As mentioned in Equations (1)–(3), $\alpha$ is regarded as a hyperparameter that adjusts the function proportion based on regional loss and distance-based loss. According to the previously mentioned rules, the value of this parameter maintained at 1 during the first 400 epochs, and it gradually dropped to 0.01 during the next 1000 epochs. Considering that the models were tested on the test set for every 40 epochs, the setting of $\alpha$ is discussed based on the results of each test. Typically, the set interval of the $\alpha$ value for similar problems could be summarized according to the $\alpha$ value of the optimal model on different databases. As observed from Figure 5, the DSC coefficients of the joint training model were improved on the two databases with the decrease in the $\alpha$ value. The best results for each model were obtained at the $\alpha$ value of 0.4–0.6. However, as $\alpha$ continues to decrease, the DSC coefficients of all models did not continue to improve, but even exhibited a downward trend in some columns.

When the value of $\alpha$ is between 1.0–0.7, the loss function is mainly based on the regional loss. As mentioned above, it may affect the training stability and performance due to the highly imbalanced data. However, when the value of $\alpha$ is less than 0.4, the loss function is dominated by the distance-based loss, which may make the function fall into a local minimum. In general, Thus, the recommended interval for the distance-based loss weight (1–$\alpha$) in the joint training strategy is 0.4–0.6.

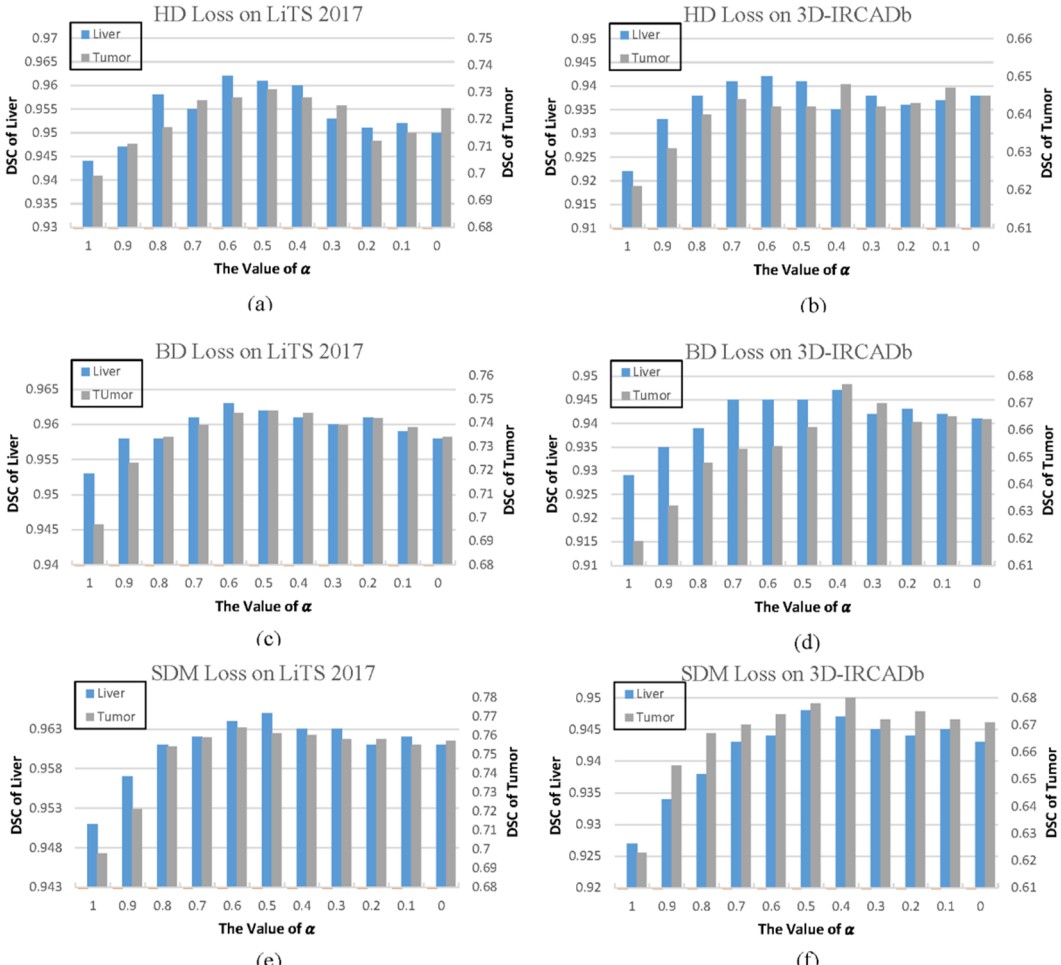

**Figure 5.** The test results on two databases with different $\alpha$ values. (**a**) The test results of models with HD loss on LiTS 2007. (**b**) The test results of models with HD loss on 3D-IRCADb. (**c**) The test results of models with BD loss on LiTS 2007. (**d**) The test results of models with BD loss on 3D-IRCADb. (**e**) The test results of models with SDM loss on LiTS 2007. (**f**) The test results of models with SDM loss on 3D-IRCADb.

Some studies point out that the earlier use of distance-based loss may lead to convergence to a local minimum or saddle point [30]. In this paper, the solution for this problem was to utilize the regional loss function only in the first period of training to avoid falling into local minima. Then, after the training entered the platform period, the distance-based loss weight gradually increased to fine-tune the results. This strategy was conceptually similar to the energy on the basis of the classical contour for the segmentation of the level set, such as the active geodesic contour [38], which also required additional regional terms to avoid trivial solutions. Taking the model based on SDM loss function as an example, according to our experimental findings in Table 4, the models utilizing the training strategy proposed in this paper achieved higher results on two databases, respectively.

A comparison of our approach with similar approaches is given in Table 5. Some values were missing because they were not available in the original article. On the LiTS 2017 dataset, compared with the other methods in the table, the algorithm in this paper obtained the best results in the liver segmentation task. It also surpassed most methods in the tumor segmentation task. Our algorithm achieved the highest DSC score, while our ASD score was slightly higher than the ASD scores in [3,45].

**Table 4.** The results on two datasets with a different training strategy.

| Training Strategy | Category | Dataset | DSC | HD95 (mm) | ASD (mm) | TNR | TPR |
|---|---|---|---|---|---|---|---|
| Joint training at the beginning | Liver | LiTS 2017 | 0.951 | 4.17 | 2.1 | 0.967 | 0.999 |
| | | 3D-IRCADb | 0.937 | 7.28 | 2.49 | 0.951 | 0.998 |
| | Tumor | LiTS 2017 | 0.742 | 8.03 | 2.38 | 0.623 | 0.999 |
| | | 3D-IRCADb | 0.633 | 12.22 | 3.55 | 0.602 | 0.999 |
| Our training strategy | Liver | LiTS 2017 | **0.965** | **3.23** | **0.88** | **0.982** | **0.999** |
| | | 3D-IRCADb | **0.948** | **4.68** | **1.81** | **0.958** | **0.998** |
| | Tumor | LiTS 2017 | **0.764** | **6.72** | **1.25** | **0.761** | **0.999** |
| | | 3D-IRCADb | **0.682** | **9.47** | **2.82** | **0.654** | **0.999** |

**Table 5.** The comparison of our approach with similar approaches.

| Approach | Dataset | Liver | | Tumor | |
|---|---|---|---|---|---|
| | | DSC | ASD (mm) | DSC | ASD (mm) |
| U-Net [39] | LiTS 2017 | - | - | 0.650 | - |
| ResNet [40] | LiTS 2017 | - | - | 0.670 | 6.66 |
| DenseNet [41] | LiTS 2017 | 0.912 | 6.49 | 0.492 | 1.44 |
| FCN+ACM [42] | LiTS 2017 | 0.943 | 2.30 | - | - |
| GIU-Net [43] | LiTS 2017 | 0.951 | 1.80 | - | - |
| ResNet [44] | LiTS 2017 | 0.959 | - | 0.500 | - |
| RA-Unet [13] | LiTS 2017 | 0.961 | 1.21 | 0.595 | 1.29 |
| H-DenseUNet [3] | LiTS 2017 | 0.961 | 1.45 | 0.722 | **1.10** |
| CDNN [45] | LiTS 2017 | 0.963 | 1.10 | 0.657 | 1.15 |
| ours | LiTS 2017 | **0.965** | **0.88** | **0.764** | 1.25 |
| MPAM [46] | 3D-IRCADb | - | 2.24 | - | - |
| ASM [47] | 3D-IRCADb | - | 1.66 | - | - |
| U-Net [3] | 3D-IRCADb | 0.923 | 4.33 | 0.510 | 11.11 |
| ResNet [3] | 3D-IRCADb | 0.938 | 3.91 | 0.600 | 6.36 |
| CFCNs [48] | 3D-IRCADb | 0.943 | **1.50** | 0.560 | - |
| H-DenseUNet [3] | 3D-IRCADb | 0.947 | 4.06 | 0.650 | 5.29 |
| ours | 3D-IRCADb | **0.948** | 1.81 | **0.682** | **2.82** |

Note: - denotes the result is not reported.

On the 3D-IRCADb dataset, the algorithm in this paper obtained the highest DSC score in the liver segmentation task, and the ASD score was only higher than the results in [47,48]. As for the tumor segmentation task, our method obtained the best results on the two indicators of DSC and ASD. As such, it can be concluded that the overall performance of our algorithm outperforms other similar algorithms in the table on the two databases.

Spatial information was not used in models based on regional metrics, and the prediction errors were treated equally. This meant that voxel errors in objects that had already been detected were as important as errors that occurred within objects that had missed totally. By contrast, because the distance-based loss was on the basis of a distance map relative to the true boundary, such cases were penalized, thus assisting in recovering the far and small regions. Therefore, our algorithm will have advantages in the tasks which need to segment a large number of small objects.

It is worth noting that for the models jointly trained with two or more loss functions, it was generally necessary to discuss the weight of each loss function. Our experimental results also showed that as the weight of the distance-based loss function increases, the performance did not continue to improve, and even showed a downward trend, which required special attention during the model training stage. In addition, it was also important that the distance-based loss function was gradually

added for joint training, only after the performance of the model on the validation set entered the platform period.

## 5. Conclusions

The present study aims to solve the problem of decreased segmentation performance of the liver and tumor segmentation algorithm due to the highly imbalanced number of voxels in the foreground and background. In this paper, an improved V-net algorithm based on region and distance metrics is applied for the 3D liver segmentation task. Three distance-based loss functions are introduced to jointly train the model with the original regional loss function, which improves the training effect and stability. Comparative experiments on the LiTS 2017 and 3D-IRCADb databases indicated the effectiveness of the improvement. Additionally, the optimal weight coefficient value for joint training is also discussed, and a new training strategy is proposed. Certainly, our findings in the present study shed new light on the solution to specific liver and tumor segmentation tasks.

**Author Contributions:** Conceptualization, Y.Z., C.L. and T.W.; Data curation, Y.Z.; Formal analysis, Y.Z.; Funding acquisition, T.W.; Investigation, Y.Z. and X.P.; Methodology, C.L. and T.W.; Project administration, T.W.; Resources, T.W.; Software, X.P. and C.L.; Supervision, T.W.; Validation, Y.Z. and X.P.; Visualization, X.P. and C.L.; Writing—original draft, Y.Z.; Writing—review & editing, Y.Z. All authors have read and agreed to the published version of the manuscript.

**Funding:** This research was funded by National Science and Technology Major Project, grant number No. 2018ZX10301201 and National Natural Science Foundation Project, grant number No. 61971445.

**Conflicts of Interest:** The authors declare no conflict of interest.

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
