# Peer review of "3D Liver and Tumor Segmentation with CNNs Based on Region and Distance Metrics"

_applsci, doi:10.3390/app10113794_

Round 1
Reviewer 1 Report
The manuscript generally well addressed about the comments. A tumor segmentation results and comparison with previous studies are conducted. While the manuscript is revised, the discussion is not enough to support the results.
- In line 97-101, please remove this paragraph.
- Section 3.2.3 Comparison and discussion, there is no discussion about Table 5. Table 5 showed that proposed algorithm showed improved result. Author should mention why proposed result showed improve result, what is the limitation and possible solution.
Author Response
The manuscript generally well addressed about the comments. A tumor segmentation results and comparison with previous studies are conducted. While the manuscript is revised, the discussion is not enough to support the results.
Response: Thanks for your advice. According to your suggestions, we have thoroughly updated the content of the study. The modifications and additions in the revised manuscript are listed as follows.
1.In line 97-101, please remove this paragraph.
Response: Thank you for your suggestion. This paragraph was removed in the revised manuscript.
2.Section 3.2.3 Comparison and discussion, there is no discussion about Table 5. Table 5 showed that proposed algorithm showed improved result. Author should mention why proposed result showed improve result, what is the limitation and possible solution.
Response: Thanks for the recommendations as well as the constructive advice. The further discussion was given in the revised manuscript. We explained why we got better results along with the limitation and possible solution. (Line392-403)
Special thanks to you for your good comments.

Reviewer 2 Report
This article presents a very interesting and important medical subject and it is worth publishing after some corrections. The article is very well structured, presents a new algorithm and, additionally, contains a completed comparison with other methods published so far. One of the novelties worth noticing is the introduction of a changing α coefficient and a very careful analysis of its influence on the results obtained.
Nonetheless, the aspect of dimensionality is unclear to me. In the title we have ‘3D segmentation’, in the main text pixels and voxels are used interchangeably. The architecture scheme in Fig. 2 does not explain if the V-Net is 2D or 3D. In line 197 the authors write about employing ‘spherical/circular convolution kernels’ which is also confusing. From Fig. 1 I conclude that the authors analyse 2D images from Fig 3, where we can see the 3 cross-sections (x,y,z) based on which a 3D model of a segmented liver is reconstructed. However, it is unclear if the authors have used 2D images as network segmentation results or, based on 2D segmentation, constructed a 3D model for the final users. If you really use the 3D V-Net, why is not the article:
‘3D U-Net: Learning Dense Volumetric Segmentation from Sparse Annotation’ by
Özgün Çiçek, Ahmed Abdulkadir, Soeren S. Lienkamp, Thomas Brox, Olaf Ronneberger
mentioned in the references?
This ambiguity has to be clearly explained to the readers
Detailed remarks:
- All abbreviations should be explained
- the ‘Euclidean distance’ is symmetric in L2 (line 178)
- Fig. 2 is hard to make out, the font is too small
- Both ‘pixel’ (line 150) and ‘voxel’ (line 153) are used to explain eq. (5), this should be clarified
- In eq. (5) ‘p’ signifies probability, in eq. (6) it means volume apparently, whereas in eq. (8) ‘p’ is a point. In one text symbols should be unequivocally defined and associated with individual, fixed designates.
- In line 174 should be ‘[29], Kervadec’
- Eq. (8) is rewritten from Kervadec’s article, but the symbols and comments should correlate with this text.
- In line 186, we can read: ‘images of brain lesion in this study’, so in this study or in reference [30]?
- What are ‘x’ and ‘y’ in eq. (11) – line 221?
- Line 229 ‘Where ?1 is the difference in ?1’ maybe LSDM?
- English should be proofread
Author Response
This article presents a very interesting and important medical subject and it is worth publishing after some corrections. The article is very well structured, presents a new algorithm and, additionally, contains a completed comparison with other methods published so far. One of the novelties worth noticing is the introduction of a changing α coefficient and a very careful analysis of its influence on the results obtained.
Nonetheless, the aspect of dimensionality is unclear to me. In the title we have ‘3D segmentation’, in the main text pixels and voxels are used interchangeably. The architecture scheme in Fig. 2 does not explain if the V-Net is 2D or 3D. In line 197 the authors write about employing ‘spherical/circular convolution kernels’ which is also confusing. From Fig. 1 I conclude that the authors analyse 2D images from Fig 3, where we can see the 3 cross-sections (x,y,z) based on which a 3D model of a segmented liver is reconstructed. However, it is unclear if the authors have used 2D images as network segmentation results or, based on 2D segmentation, constructed a 3D model for the final users. If you really use the 3D V-Net, why is not the article:
‘3D U-Net: Learning Dense Volumetric Segmentation from Sparse Annotation’ by Özgün Çiçek, Ahmed Abdulkadir, Soeren S. Lienkamp, Thomas Brox, Olaf Ronneberger mentioned in the references? This ambiguity has to be clearly explained to the readers
Response: Thanks for the recommendations as well as the constructive advice. According to your suggestions, we have thoroughly updated the content of the study. We declared that the input data was 3D data in the revised manuscript. (Line 103-108, Line 134, Line 235, Line 242, Line 248……) In response to this question, the modifications and additions in the revised manuscript are listed as follows.
- For more rigorous expression, all ‘pixels’ were replaced with ‘voxels’ in the revised manuscript.
- The ‘spherical/circular convolution kernels’ was replaced by ‘spherical convolution kernels’ for the data format was three-dimensional.
- We have updated Fig.1 and Fig.3 and we made corrections throughout the manuscript. The data diagram in the Fig.1 and Fig.3 is replaced with the rendered 3D data.
- Thanks for the advice. The article which you recommended was important. We added the description part in the revised manuscript “Özgün Çiçek et al. introduced a 3D U-Net network for volumetric segmentation that learns from sparsely annotated volumetric images [28]. The network extended the previous U-Net [18] architecture by replacing all 2D operations with their 3D counterparts. The implementation performed on-the-fly elastic deformations for efficient data augmentation during training.” (Line70-73)
Detailed remarks:
- All abbreviations should be explained
Response: Thanks for the advice. We made corrections throughout the manuscript.
- the ‘Euclidean distance’ is symmetric in L2 (line 178)
Response: We are very sorry for our negligence. We made corrections in the revised manuscript. “A symmetric L2 distance (Euclidean distance) on the space of shapes… ” (Line177-178)
- Fig. 2 is hard to make out, the font is too small
Response: Many thanks for your careful review. We have updated Fig.2 in the revised manuscript.
- Both ‘pixel’ (line 150) and ‘voxel’ (line 153) are used to explain eq. (5), this should be clarified
Response: Thanks for your careful review. All of the ‘pixels’ were replaced with ‘voxels’ in the revised manuscript.
- In eq. (5) ‘p’ signifies probability, in eq. (6) it means volume apparently, whereas in eq. (8) ‘p’ is a point. In one text symbols should be unequivocally defined and associated with individual, fixed designates.
Response: Thanks for the advice. We made corrections to make sure that all of the symbols the revised manuscript were unequivocally defined.
- In line 174 should be ‘[29], Kervadec’
Response: We are very sorry for our negligence. We made corrections in the revised manuscript. (Line174)
- Eq. (8) is rewritten from Kervadec’s article, but the symbols and comments should correlate with this text.
Response: Thanks for the advice. We revised it in the revised manuscript. (Line182)
- In line 186, we can read: ‘images of brain lesion in this study’, so in this study or in reference [30]?
Response: We are very sorry for our negligence. We made corrections in the revised manuscript. “The boundary loss function was used to segment the Magnetic Resonance (MR) images of brain lesion in [30], and Dice and Hausdorff score increased by 8% and 10%” (Line184-186)
- What are ‘x’ and ‘y’ in eq. (11) – line 221?
Response: Thank you for your suggestion. We have supplemented the explanation in the revised manuscript. “For the target organ as well as a point x shown on 3D medical image, y is the most adjacent point on the target organ surface” (Line218-219)
- Line 229 ‘Where ?1 is the difference in ?1’ maybe LSDM?
Response: We are very sorry for our incorrect writing. We made corrections in the revised manuscript. “Where represented the loss which is the difference between the predicted and the real SDM values.” (Line228-229)
- English should be proofread
Response: Thank you for your suggestion. We conducted a further examination of the language expression of the revised manuscript.
Special thanks to you for your good comments.

This manuscript is a resubmission of an earlier submission. The following is a list of the peer review reports and author responses from that submission.
Round 1
Reviewer 1 Report
The study report on improvement of 3D liver segmentation with CNN in conjunction with various loss functions. The authors utilized three different loss function based on distance metrics. The major contribution of this work is the distance metrics based on loss function improve the liver segmentation. While the manuscript showed interesting results, it lacks some significant information in the material and method, the description about major results are not clear, and the discussion is not enough to support the results. My overall recommendation for the manuscript is to revise the manuscript and address the following points, before it can be published in applied sciences.
- Two datasets were used for this study which has liver and tumor regions. The results were segment only normal area, not tumor area? Were any normal liver images used for this study?
- This study combined previous technique which is already published. Please do comparison with previous studies. And please add the result such as ROC curve, specificity and sensitivity.
- Figure 2 showed 3D-IRCADb datasets as an example. Please show the LiTS 2017 datasets. Please check the data name 3D-IRCADb or 3Dircadb.
- In section 3.2.1, “In the LiTS 2017 data set, 131 scans were used as the experimental data, among which, 102 were used for training, 20 for verification, and 9 for testing. In the 3Dircadb data set, 20 scans were selected, including 10 utilized as the training set, 5 as the validation set, and 5 as the test set.” Any cross validation was conducted? For example, CNN was trained using LiTS 2017 datasets and tested using 3Dircadb or vice versa.
- Based on Table 1, 3Dircadb showed much more difference between LDice and other loss function in 95th-percentile of and ASD than LiTs 2017. Please describe why this difference was occurred.
- 3Dircadb has small data number. Does it affect the result?
- Please check the reference format. Every reference has et al after the author name and applied science use abbreviation in journal name.
- In line 22, please use round brackets in SDM.
- In line 57 - 62, please add the description about multi-channel FCN.
- In line 75-76, please add description about Hausdorff loss and Signed Distace Map.
- Please clearly describe line 83-84.
- Please change Figure 1 to readable figure.
- In equation (1) and (2), there is only denote i and j. Please change the equation for 3D data.
- In line 125, please describe what does L2 mean.
- In line 130-133, please add reference.
- In equation (4) please clarify about lambda.
- In equation (5) please clarify about omega.
- In 167-168, only (5) was used in this study. Please remove equation (6), (7) and (8).
- In line 175, please clarify about SDM. Signed distance map or symbol distance map?
- In line 179-180, please describe what point x, R3 and R mean?
- In line 190-200, Please move to this paragraph. It is not suitable for this section.
- In equation (11), Please clarify about Lossseg.
- In line 196-197, “the boundary loss function was used to assist the original loss function to fine-tune the 196 training models” Please describe why boundary loss function was used instead of HD or SDM.
- In line 203, “We conducted evaluation on three datasets for medical image segmentation.” In this study, two datasets were used. Please clarify it.
- In line 213, remove “in”.
- In line 216, “We trained the three 3D V-Net models with three different loss functions in two databases.” In the result, four Loss functions were showed (LossDice, LossHD, LossBD, LossSDM). Please clarify how many V-net was trained.
- Please use consistent term for Lossoriginal and LossDice.
- In line 233-234, Please clarify what do the DSC, HD and ASD values change mean.
- In table 1, please use same digit numbers after the decimal point.
- In line241, please clarify about WMH dataset and check the value 1.2%.
- In line 243-244, please clarify the value increase or decrease from which parameter to which parameter and check the value 53.7%.
- In figure 3 caption, please check the green. Only red and blue mark in the figures.
- In line 260-261, please show which image is imbalanced image as an example in figure 3.
- In line 276-277, the figure 4 not decreased in all panels. Please clarify it.
- In line 277-278, “the best results were 277 obtained for each model at the ? value of 0.4-0.6.” Please clearly describe how this value considered as best results.
- In line 291, some numbers are higher in "Our training strategy at HD and ASD". Please clarify it.
Reviewer 2 Report
The main contribution of the paper is that existing V-net algorithm (2016) was combined with recently introduced metrics (2018, 2019 and 2019). It was tested on two datasets (LiTS 2017 and 3Dircadb). Dice coefficients were improved by 1.2% and 1.7% respectively. Another contribution of the paper is also a training strategy, which improved results.
Disadvantage of the paper is, that the data were down-sampled to 50% of the original data. Although the results are very interesting, contribution of the paper submitted to a journal with IF 2.217 is not sufficient and it does not stand out from the average conference papers. I also disagree with the claim of the authors that they designed an algorithm for regional and boundary losses. Algorithm and metrics already exist, are cited - the unique in this paper is using them together with V-NET.
Comparison with the state-of-the-art is not sufficient.
Figure 1 was taken from the original paper [29]. Although the original paper is cited in the text, the figure is presented as to be a contribution of the paper. There is not declared under which license the paper was taken and is potential plagiarism: https://arxiv.org/pdf/1606.04797.pdf
Formal notes:
PReLu -> PReLU
"in in inhaled"
Variables should be in italics according to mathematical standards (N, j, S, G, …)